# The Role of Leukemia Inhibitory Factor in Counteracting the Immunopathology of Acute and Chronic Lung Inflammatory Diseases

**Howard Yu** [1], **Sahil Zaveri** [1], **Meshach Pillai** [1], **Harsha Taluru** [1], **Michael Schaible** [1], **Sahil Chaddha** [1], **Asad Ahmed** [1], **Said Tfaili** [2] **and Patrick Geraghty** [1,*]

[1] Department of Medicine, State University of New York Downstate Health Sciences University, 450 Clarkson Avenue, Brooklyn, NY 11203, USA; howard.yu@downstate.edu (H.Y.); sahil.zaveri@downstate.edu (S.Z.); meshach.pillai@downstate.edu (M.P.); harshavardhan.taluru@downstate.edu (H.T.); michael.schaible@downstate.edu (M.S.); sahil.chaddha@downstate.edu (S.C.); asad.ahmed@downstate.edu (A.A.)

[2] Department of Pediatrics, University of Wisconsin School of Medicine and Public Health, 600 Highland Avenue, Madison, WI 53792, USA; saidtfaili@yahoo.com

[*] Correspondence: patrick.geraghty@downstate.edu; Tel.: +1-718-270-3141

**Abstract:** Leukemia inhibitory factor (LIF), a member of the IL-6 cytokine family, is highly expressed throughout the body in multiple tissues and cell types. LIF is primarily known to induce the differentiation of myeloid leukemia cells, but recent studies show that LIF has many other functions, including playing multiple roles in cancer and normal physiology. LIF expression is linked to cellular proliferation, metastasis, inflammation, and chemoresistance. LIF expression and secretion are triggered by many means and its downstream signaling can vary based on tissue types. Recent publications suggest that LIF may play a role in pulmonary diseases and its regulation is altered through external factors, such as cigarette smoke, inflammation stimuli, or infections. This review outlines the current knowledge of the function of LIF protein, mediators of LIF expression, receptors it interacts with, downstream LIF signaling, and possible pulmonary outcomes mediated by LIF.

**Keywords:** leukemia inhibitory factor; respiratory syncytial virus; chronic obstructive pulmonary disease; STAT3; surfactant protein; foxp3; regulatory T cell

## 1. Introduction

The leukemia inhibitory factor (LIF) protein was initially discovered in 1988 as a glycoprotein that was observed to be released by ascites tumor cells. LIF mediates various biological functions but one of LIF's most remarkable biological effects is its activity on murine embryonic stem cells [1–3]. It exhibits a distinct behavior from M1 cells as it suppresses the differentiation of embryonic stem cells in vitro [4,5]. Embryonic stem cells cultured with purified recombinant LIF preserve their pluripotent state through multiple passages [6]. LIF expression, however, goes beyond myeloid and embryonic stem cell lines. Owing to its pleiotropic effects, LIF is expressed in several tissues, including the kidney, olfactory neuronal tissue, lung, heart muscle, and brain [7,8]. LIF signaling is regarded as playing a significant role in many processes, including proliferation and differentiation of leukemia and hemopoietic cells, pluripotent stem cell self-renewal, tissue/organ development and regeneration, neurogenesis and neural regeneration, maternal reproduction, immune response, and metabolism.

LIF overexpression is seen in solid tumors, including cholangiocarcinoma, breast, colorectal, and pancreatic cancer, despite its ability to suppress murine myeloid leukemia. According to research by Wu et al., hypoxic conditions within a solid tumor microenvironment stabilize hypoxia-inducible factor-2 (HIF-2), which causes it to bind to the LIF promoter and induce the expression of LIF [9]. Correspondingly, induction of perinatal

cerebral hypoxia-ischemia in the Wistar mouse model resulted in LIF being produced from astrocytes in the subventricular zone. This led to neurogenesis and regeneration, demonstrated by clonal expansion of neural stem and progenitor cells, presumably via the Notch-Delta/Serrate/Lag2 (DSL) signaling pathway [10].

Evidence of LIF involvement in multiple tissue types and disease states emphasizes the context-dependent nature of its function. Notably, research involving the role of LIF in pulmonary tissue, especially its role in acute and chronic lung diseases such as COPD and emphysema, is sparse. Several studies demonstrate that LIF protects the lung from a respiratory syncytial virus (RSV) infection injury and intra-tracheal lipopolysaccharide (LPS) challenge in the murine model [11,12]. Overexpression of LIF in airway epithelial cells protects the airways during hyperoxia in mice, with decreased pulmonary edema and improved survival [13]. LIF is a prominent activator of signal transducer and activator of transcription 3 (STAT3) [14]. Loss of STAT3 correlates with significantly greater inflammatory proteins detected in bronchoalveolar lavage fluid (BALF). An increase in interleukin (IL)-1α and monocyte chemotactic protein (MCP)-1 correlates with elevated frequency of BALF macrophages, a key player in inducing proteases leading to characteristic dysfunctional airway remodeling seen in chronic obstructive lung diseases [15]. This highlights LIF's potential role in moderating inflammatory cytokines expression and altering alveolar monocyte response to acute injury. It is also important to note that LIF and the LIF-receptor (LIFR) α are constitutively expressed during fetal lung development and they play an inhibitory physiological role on fetal lung branching [16]. Equally, LIF may play a role in other factors associated with disease initiation and progression, such as the regulation of cell differentiation, neuronal development, and stem cell self-renewal. Here, we review the current knowledge of the function of LIF protein, mediators of LIF expression, receptors it interacts with, downstream LIF signaling, and possible pulmonary outcomes mediated by LIF.

## 2. LIF Gene Regulation

Early molecular biology studies demonstrated high LIF gene sequence fidelity across four mammalian species: murine, human, porcine, and bovine. This homology is especially true within the four TATA box elements of the promoter region, as well as candidate repressor and non-coding control sequences [17]. This observation suggests that LIF's vital role in cell signaling is not species dependent. The LIF gene maps to 22q11–q12.2 and spans 76 kb [18]. It is transcribed in a telomeric to centromeric fashion with a posttranscriptional product containing three exons [19]. Methylation of DNA at position 5 cytosine within a CpG dinucleotide is one of the predominant means of epigenetic modification in eukaryotes. Within the promoter sequence, methylation of CpG islands negates transcription due to inhibiting transcription factor binding [20]. Recent studies regarding LIF gene regulation via epigenetic modification were performed by investigating the role of LIF overexpression in breast cancer tumorigenesis. By examining normal MCF-10A epithelial cells and their cancerous variants, it was identified that CpG regions exist within the promoter and first intron of the LIF gene [21]. Furthermore, it was demonstrated that 9 of 16 CpG pairs (CpG 1–9) in the promoter region between nucleotides −631 and −335 were heavily methylated, thus corresponding to reduced LIF expression in noncancerous epithelial cells [21].

In 1999, Haines and Voyle discovered variability in expressed LIF transcripts [22,23]. It was later determined that expression of the LIF gene results in three independently functioning LIF transcripts: LIF-D, LIF-M, and LIF-T [22,23]. Mouse transcriptome analysis revealed high basal LIF-D and LIF-M expression levels relative to LIF-T [22,23]. Differing LIF isoforms are generated via an alternatively spliced first exon onto a common second and third exon [22,23]. The first exon of LIF-D and LIF-M contains an ATG start codon, leading to the expression of a secretion peptide sequence [22,23]. Contrastingly, the first exon of LIF-T does not contain an ATG sequence; this leads to its N-terminal truncation by

22 peptides and its exclusively intracellular activity [22,23]. Abrogation of LIF-T isoform activity with serine protease inhibitor CrmA, but not Bcl-2, suggests interaction with intracellular caspase-activating pathways [24]. Overexpression of LIF-T in COS-1 fibroblast-like cells and 293 T kidney epithelial cells leads to apoptosis. However, whether LIF-T directly induces apoptosis or is a by-product of other apoptotic signaling pathways remains to be determined [24]. LIF expression is triggered by several external factors, such as TGFβ [25] and estrogen [26]. Therefore, LIF has variable transcripts, isoforms, and stimuli which regulate its expression and subsequent downstream signaling.

## 3. LIF Protein Structure

Post-translationally, the 202 amino acids (AA) LIF peptide is cleaved into a 20 kDa gene product [27]. Early work on solving LIF protein structure by using X-ray crystallography and nuclear magnetic resonance (NMR) spectrometry [28,29] showed that the LIF protein consists of a four α-helix bundle, with the helices arranged in an up-up-down-down left-handed fashion [30]. The first helix, Helix A, begins at Leu44 (residue 22 of the mature chain). It is bonded to the C-terminal of Helix C via two disulfide bonds (Cys34–Cys156 and Cys40–Cys153). A third disulfide bond joins helix D to the linker between helices A–B [30]. The N-terminal "flap" preceding helix A is tethered at both ends by disulfide bonds (Cys12–Cys134 and Cys18–Cys131) and participates in receptor binding [30]. These structural modifications influence the receptor binding potential of LIF.

### Receptor Binding

LIF signaling competent complex consists of three subunits: LIF ligand, LIF receptor, and a gp130 co-receptor [31]. LIF-receptor (LIFR) and gp130 share similar structural homologies as both have JAK tyrosine-kinase membrane-bound via their intracellular domain [32]. LIF helices A, C, and the N-terminal flap that precedes helix A serve as the ligand in interaction with the D2 domain of gp130 [30]. LIF avidly binds to both gp130 and LIFR with high affinity, with the interaction of LIF/LIFR (Kd = 1 nM) being approximately 80-fold tighter than of LIF/gp130 (Kd = 80 nM) [33]. This is understandable, given gp130's cross-reactivity in its ability to bind a diverse group of hematopoietic cytokines [30,33]. Mature LIF is heavily N-glycosylated, with an apparent molecular mass range of 32–62 kDa in the glycosylated state. However, glycosylation is not essential for its activity [1,29,34]. Once activated, receptor-bound kinases auto-phosphorylate the LIFR and gp130 dimer, which serves as a recruitment site for canonical secondary signaling molecules such as STAT3, PI(3)-kinase, and MAPK [32,35,36].

## 4. LIF Downstream Pathways
### 4.1. STAT3

Once activated by LIF, receptor-bound JAK1 initially targets five tyrosine domains on the intracellular regions of the LIFR:gp130 dimer for phosphorylation. These serve as binding sites for STAT3, PI(3)-Kinase, and MAPK [27,37]. STAT proteins are a family of transcription factors located within the cytoplasm [38]. Of these, STAT3 is the most critical transducer of the LIF-activated receptor complex [39,40]. Once phosphorylated, STAT3 translocates to the nucleus, where it functions to upregulate cytokine production [41]. Autoregulation of the LIF/STAT3 axis involves a negative feedback mechanism via the suppressor of cytokine signaling-3 (SOC3) [42–44]. Increased SOC3 activation suppresses LIF/STAT3 response via ubiquitination of phosphor-tyrosine motifs on gp130, thereby directly counteracting JAK kinase [45]. Of note, SOC3 also competitively inhibits SHP2, a mediator of the LIF/MAPK signaling cascade, thus acting as a regulator of LIF/MAPK signaling [46].

STAT3 activation is critical in maintaining embryonic stem cells in their undifferentiated state [47]. The proposed mechanism for this effect is possibly linked to STAT3-induced transcription of crucial cell cycle regulator genes *Myc* and *Nanog* [47]. The effect of STAT3 on cell differentiation suggests a possible mechanism behind LIF's classic ability to in-

hibit differentiation while promoting self-renewal. This further highlights LIF as a possible player in maintaining respiratory epithelial resilience and renewal in both acute and chronic inflammatory lung diseases [15,48].

### 4.2. PI(3)Kinase Signaling

Another mechanism for immediate LIF signaling relays is via protein PI(3)Kinase, a catalytic enzyme that produces the secondary signaling molecule phosphatidylinositol (3,4,5)-triphosphate [36]. Ligand-bound LIF receptor subunits phosphorylate the p85 subunit of PI(3)Kinase leading to the activation of protein kinase B, a membrane-bound signal transducer molecule, implicated in the activation of a variety of intracellular pathways, including mTOR and Wnt signaling [49]. Paling et al. elucidated the significance of PI(3)Kinase signaling by demonstrating that embryonic stem cells can maintain an undifferentiated state using artificially enhanced protein kinase B expression in the absence of LIF stimulation [50]. It should be noted that all three pathways (PI(3)Kinase, mTOR, and Wnt) are linked to playing major roles in lung development, lung repair, remodeling, and regeneration. Dysfunctional repair, remodeling and regeneration is linked to the progression of many lung diseases [51–53]. Determining the involvement of LIF in these processes would greatly enhance our understanding of multiple disease associated pathways.

### 4.3. MAPK Signaling

MAPK signaling proceeds after LIF stimulation via the recruitment of SHP2 to the phosphorylated LIFR [54]. Activated SHP2 activates MAPK via Ras/Raf signaling. Activation of MAPK must be taken in the context of JAK/STAT co-activation [55]. Independent activation of MAPK leads to a loss of the stem-cell state in the absence of STAT3 co-stimulation [56,57]. However, once ligand bound, LIFR/gp130 co-stimulation of STAT3 (to a greater degree as compared to MAPK) leads to competitive inhibition of SHP2 via SOCS3, which results in MAPK inhibition [58]. LIF significantly inhibits lung growth in rat fetal lung explants, but increases in the MAPK, p44/42, phosphorylation, while inhibition of LIF significantly stimulated lung growth via another MAPK, p38, and Akt signaling [16]. Therefore, LIF induction of STAT3/gp130, PI(3)Kinase, and MAPK could regulate many processes in lung diseases.

## 5. Regulators of LIF Gene Expression

### 5.1. Hypoxia-Inducible Factors

LIF expression can be induced in response to many extracellular and intracellular conditions, including hypoxia, inflammation, cell injury, and tissue invasion in various cancer states. Hypoxic conditions allow for the stabilization of hypoxia-inducible factors (HIFs) [59]. HIFs are heterodimeric transcription factor proteins consisting of an $\alpha$- and $\beta$-subunit [59]. The binding of HIFs to specific hypoxic response elements (HREs) DNA sequences allows cells to modify gene transcription in hypoxic environments [59]. By the use of luciferase reporter vectors containing the HRE site, Wu et al. demonstrated in colorectal and colonic carcinoma cell lines that two HREs promoter regions occurring upstream to exon 1 of LIF gene: segments HRE-D ($-2791$ bp) and HRE-C ($-1769$ bp) are bound via HIF-$2\alpha$ but not HIF-$1\alpha$ after 36 h of hypoxic culture conditions. This led to subsequent induction of LIF transcription and possibly increased chemo-resistance [9].

### 5.2. IL-1β and IL-6

Intratracheal delivery of LPS induces LIF secretion into the alveolar space in mice [11]. Interestingly, mice co-injected with LIF and LPS demonstrated markedly reduced TNF-$\alpha$ levels compared to controls [11]. Human fetal lung fibroblast, bronchial epithelia, and bronchial smooth-muscle cells treated with IL-1β induced LIF production within 1–2 h of stimulation exhibiting increasing levels of LIF in a time-dependent fashion up to 48 h post-challenge [60]. Treatment with IL-6 yielded statistically significant albeit reduced LIF expression relative to IL-1β. Cells stimulated with LIF did not yield substantial LIF

expression, arguing against LIF self-induction [60]. Therefore, LPS, IL-1β, and IL-6 can trigger LIF expression within the lungs.

### 5.3. TGF-β

Multiple studies have reported the induction of LIF mRNA and protein by TGF-β in Schwann, glioblastoma, and melanoma cell models [61,62]. By cloning the LIF promoter sequence within luciferase reporter constructs, Penuelas et al. could trace the TGF-β responsive element of the LIF promoter to the −276/−73 bp region, which contained a single Smad-binding element [62]. siRNA knockdowns of Smad-2, -3, and -4 in glioblastoma astrocytoma cell lines demonstrated decreased LIF mRNA expression in the presence of TGF-β [62]. Taken together, induction of LIF by TGF-β is mediated in a Smad-dependent manner [62].

The role of LIF induction in lung inflammation may not be limited to IL-1β, IL-6, or TGF-β alone. Early experiments from Wetzler et al. demonstrated increased levels of LIF RNA following exposure of normal human bone marrow stromal cells to a host of other inflammatory modulators, including IL-1α, IL-1β, and TGF-β [63]. This would suggest that LIF may be responsive to a broad range of inflammatory cytokines. It is important to note that both Wetzler et al. and Carlson et al. demonstrated a basal expression of the LIF gene in bone marrow stromal cells and superior cervical ganglion cells, although tissue-specific and systemic LIF protein levels remain undetectable [63,64]. This could suggest that in basal conditions, transcription of LIF is constitutive, however, post-transcriptional RNA is degraded rather than processed. The relationship between pro-inflammatory cytokine induction of LIF expression in human lung tissue is correlated clinically by the observation of increased LIF in BALF of patients with acute respiratory distress syndrome (ARDS) [65]. However, the direct impact of TGFβ-induced LIF expression and subsequent downstream signaling, such as possible epithelial-mesenchymal transition (EMT) or fibrotic responses, require further investigation.

### 5.4. NF-κB RelA

Traber et al. identified NF-κB RelA, a prominent player in promoting innate lung immunity, as an inducer of LIF expression in lung epithelium, specifically alveolar type II cells [66]. They determined that activation of NF-κB RelA in myeloid cells promotes epithelial LIF responses during lung infections such as pneumococcal pneumonia. This induction of LIF promotes pulmonary homeostasis in response to infection [67].

### 5.5. P53

LIF expression is regulated by tumor suppressor p53 in the H1299 human non-small cell lung carcinoma cell line derived from the lymph node [68]. Hu et al. located consensus p53-binding elements within the first intron (bp +873–+898) of the LIF gene. Stimulation of H1299 cells with endogenous p53 triggered a 7-fold increase in LIF mRNA expression and a 16-fold increase in LIF protein levels within the culture supernatant [68]. This p53 regulation of LIF expression is observed to play a role in maternal reproduction [68], but has yet to be explored within the lungs.

### 5.6. Activating Transcription Factor (AT) 3

The research in human endometrial stromal cells, placental epithelial cells, and endometrial adenocarcinomatous cells demonstrates up-regulation of the LIF gene in the presence of activating transcription factor 3 via binding of consensus DNA sequence (TGACGTCA) located within 1331 to 1181 bp region of LIF gene promoter [69]. This LIF induction by ATF3 was observed in embryo development [69] but lung studies are lacking.

### 5.7. TLR5

Stimulating TLR5 with flagellin in 85As2 cells derived from human gastric cancer demonstrated increased LIF production via IL-1 receptor-associated kinase4 expression.

This led to worsening cachexia in rats [70]. Thus far, this is the only study linking TLR5 signaling to LIF expression, but this may be important in acute lung infections with pathogens such as *Pseudomonas aeruginosa* [71]. TLR5 stimulation by its ligand, flagellin, can enhance protection against *Pseudomonas aeruginosa* [71]. However, LIFs role has not been investigated in TLR5-mediated protection.

## 6. LIF Signaling in Pulmonary Diseases

### 6.1. Non-Small Cell Lung Cancer (NSCLC)

LIF expression correlates with aggressive tumor growth in NSCLC. It is especially higher in adenocarcinoma. NSCLC-derived A549 adenocarcinoma cells treated with LIF showed an increase in proliferation and tumor cell migration [72]. Several studies suggest that LIF activation of both the JAK/STAT and PI3K/AKT pathways are associated with the promotion of tumor growth and metastasis [72]. Despite LIF levels being elevated in NSCLC, LIFR expression is downregulated by mutant KRAS in NSCLS and impacts neoplastic transformation [73]. Therefore, reduced levels of LIFR may suggest reduced receptor responses despite elevated levels of LIF. Further functional studies would be beneficial with this conflicting evidence. Unlike LIFR, STAT3 is persistently activated in over 50% of tumors from NSCLC patients [74,75]. Currently, a phase I, open-label, dose-escalation clinical trial is taking place utilizing MSC-1 (AZD0171), a humanized monoclonal antibody that binds to LIF, in a population of patients with many types of cancers [76]. The data from this trial and other studies will aid in our understanding of LIF's role in NSCLC and early data suggests changes in macrophage phenotypes following treatment [76]. LIF expression in macrophages shifts macrophages from a pro-inflammatory (M1) to an anti-inflammatory (M2) phenotype [77]. This M2 macrophage phenotype is deemed a tumor-associated macrophage [78] and could play a role in multiple pulmonary diseases, in addition to NSCLC.

In breast cancer cell lines, endogenous expression of LIF can promote EMT [79], a common feature observed in cancers but also in other pulmonary fibrotic diseases. LIF's role in EMT was STAT3- and miR21-dependent [79]. More information is required on the role of LIF and EMT, but since EMT is associated with inflammation, hypoxia, and senescence, it would be surprising if LIF did not have a role in EMT within the lungs.

### 6.2. Acute Viral Lung Injury

Foronjy et al. observed increased levels of secreted LIF in airway epithelial cells following an RSV challenge. Further data regarding LIF's role in acute inflammatory response was elucidated in vivo by administration of LIF-neutralizing IgG antibody to RSV infected mice. The mice exhibited increased airway epithelial cell apoptosis, alveolar damage, inflammatory cytokine expression, and airway hypersensitivity to methacholine challenge observed in these animals [12]. This suggests that LIF plays an essential role in protecting the lung from RSV infectious injury while being a key mediator in airway response during the acute inflammatory period.

### 6.3. Acute COPD Exacerbation

Pre-existing chronic lung disease correlates with worsening RSV disease course [80]. Foronjy et al. demonstrated enhanced airway damage observed in smoke-exposed C57BL/6J mice infected with RSV [81]. Poon et al. observed reduced LIF protein levels in the BALF from COPD patients alongside increased mRNA and protein degradation in the epithelial cells isolated from COPD patients, and subsequently infected with RSV in vitro. While serine proteases are a primary driver in LIF/LIFR protein degradation, the mechanism behind LIF mRNA destabilization and increased turn-over remain to be elucidated [48]. However, these findings further highlight decreased LIF expression in the setting of chronic smoke exposure and the COPD disease state, and could represent a potential factor in viral COPD exacerbations. Interestingly, LIF expression is critical for blastocyst

implantation during pregnancy. It would be of interest to determine whether cigarette smoke exposure impacts on blastocyst implantation in a LIF-dependent manner [82].

### 6.4. Acute Respiratory Distress Syndrome (ARDS)

High levels of LIF are detected in BALF from ARDS patients [65], and the production of LIF may be an indicator of the severity of ARDS [83]. Interestingly, treatment with anti-LIF antibodies results in enhanced bacteremia and gene changes linked to cell death and epithelial homeostasis in *E. coli*-infected mice [14]. A recent bacterial pneumonia animal study using *E. coli* demonstrated that blocking LIF responses resulted in greater lung cell death and LIFR expression on epithelial cells, protecting lung tissues during pneumonia [84]. They also found that exogenous delivery of LIF significantly decreased lung injury scores in *E. coli*-infected mice [84]. A recent paper suggested that LIF should be targeted as a treatment for ARDS, with intraperitoneal LPS or liposomal delivery treatment in mice inducing Stat1(Ser727) phosphorylation, promoting interferon regulatory factor 1 (Irf1) transactivation and up-regulation of LIF which inhibited endothelial cell proliferation [85]. Since these papers utilize very different exposures to mice, additional studies are required to determine the role of LIF in ARDS and acute lung injury.

### 6.5. Idiopathic Pulmonary Fibrosis (IPF)

At the time of writing this review, there is little to no literature exploring the role of LIF in IPF. However, several recent studies demonstrate that LIF plays a role in fibrosis in other organs, such as renal interstitial fibrosis [86]. Xu et al. demonstrate that LIF is the most upregulated IL-6 family member in both human and mouse renal fibrotic lesions, and they determined that the LIF-LIFR-EGR1 axis and Sonic Hedgehog signaling influenced signaling between fibroblasts and tubular cells that promoted pro-fibrotic response via ERK and STAT3 activation [86]. Lung endothelial cells do have enhanced LIF expression after injury with bleomycin [87]. In addition, gene ontology studies indicate that LIF responses are sensitive to miR-324 signaling [88] and expression of miR-324 is downregulated in IPF patients [89]. However, this requires further investigation along with any potential role of LIF in inflammation responses, macrophage polarization, TGFβ signaling, and EMT in IPF.

### 6.6. Asthma

Serum LIF levels are higher in subjects with mild asthma than in normal subjects [90]. Higher levels of LIF are observed in obese asthmatics compared to lean asthmatic subjects [91]. A genetic study in Germany identified that the LIFR gene along with three other genes was associated with asthma symptoms [92]. LIF expression is increased in the airway epithelial tissue of ovalbumin-exposed rats [93,94] and house dust mite-treated mice [95]. Inhibition of purinergic P2Y6 receptors signaling with GC021109 reduces LIF expression and the asthmatic phenotype (including airway hyperresponsiveness, peribronchial and perivascular inflammatory cell infiltration, and α-SMA) in mice [95]. However, whether LIF was contributing to these readouts was not determined. Stimuli with LIF reduces the expression and presentation of the natural killer group 2 member D (NKG2D) receptor on natural killer cells [96]. Equally, LIF regulates $Ca^{2+}$ signaling and isometric airway contractility to acetylcholine in immature isolated rat tracheas [97]. LIF is not chemotactic for eosinophils, but augments chemotaxis, increases eosinophil peroxidase release 4-fold in vitro, and increased CD69 expression on eosinophils [90]. Therefore, sufficient data are available on LIF signaling in asthma, but functional and causality approaches are limited to determine the true role of LIF in asthma.

## 7. Immune System Interaction

### 7.1. Promoting Surfactant Synthesis

Currently, the literature is sparse on LIF's comprehensive role in innate/adaptive lung recovery. Based on current knowledge, given LIF's demonstrated role in inducing STAT3, it is plausible LIF in turn influences innate/adaptive lung recovery through its

downstream mediator STAT3. Surfactant depletion is recognized as a driver of ARDS [98]. Using hyperoxia to induce ARDS in mouse models with selective deletion of STAT3 in respiratory epithelial cells, via doxycycline-inducible Cre-recombinase gene silencing, Isamu et al. observed that STAT3-deleted mice suffered more rapidly progressive and severe lung injury [99]. Consistent with prior in vitro studies demonstrating IL-6-induced STAT3 regulation of protein surfactant B in human pulmonary adenocarcinoma, the STAT3 knockout mice demonstrated reduced or absent lipid and surfactant protein levels in BALF [100,101]. Intratracheal injection of surfactant protein B (SP-B) into these mice rescues lung damage histologically while improving the survival of mice. This suggests that the absence of STAT3 and resultant surfactant protein depletion contributes to increased lung damage [100].

There is reduced surfactant-associated saturated phosphatidylcholine (Sat-PC) in BALF after inhibiting STAT3 in lung type II alveolar pneumocytes [102]. RT-PCR analysis of type II pneumocytes showed a significant reduction in genes related to precursor uptake (Glut1, Slc34a2), surfactant synthesis, and secretion (Akt2, Srebf-1, Acox2, Glut1, Cds2, Slc34a2, Gpam) [102]. This suggests a mechanistic link between STAT3 deletion and decreased lung surfactant. Given LIF's ability to activate STAT3, we suggest a mechanism for LIF responses following acute stress in the lungs (Figure 1).

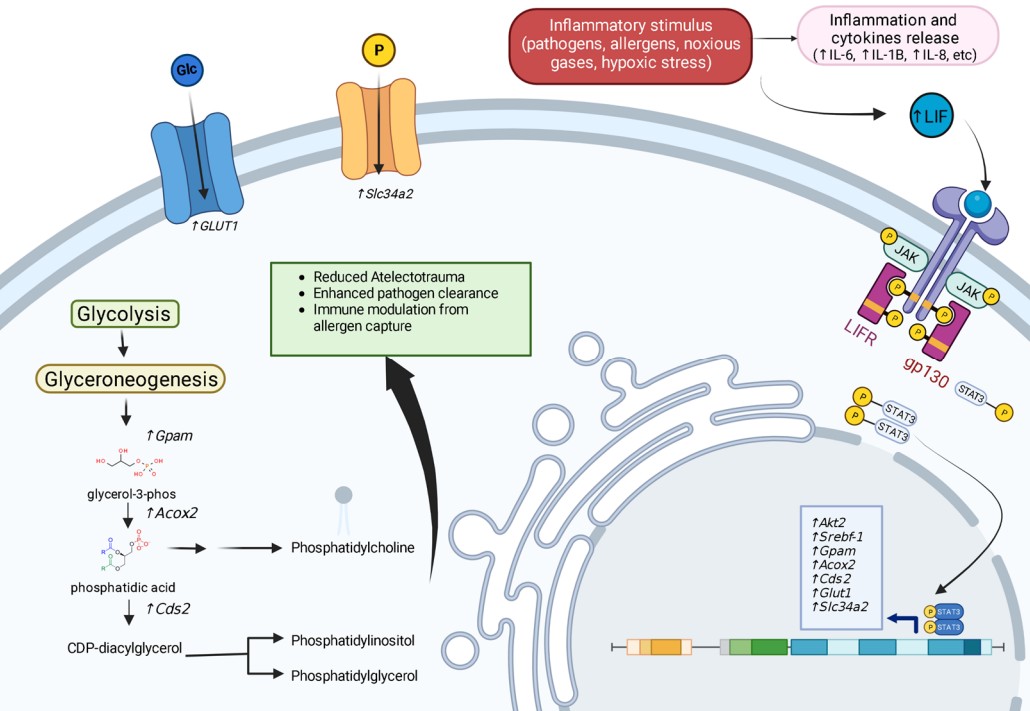

**Figure 1.** Proposed mechanism for LIF's role in assisting lung recovery in ARDS. Inflammatory stimuli such as harmful gases, pathogens, allergens, and hypoxia can trigger the release of cytokines and a compensatory response from LIF. Once bound to the LIFR/Gp130 receptor, LIF can act locally and activate gene transcription through STAT3. LIF can then increase the transcription of genes involved in glyceroneogenesis precursor uptake (Glut1, Slc34a2), steroid and fatty acid biosynthesis (Srebf-1), as well as key regulators of de novo surfactant production (Gpam, Acox2, Cds2). This results in increased surfactant production, which enhances lung resilience and recovery by reducing atelectotrauma, improving pathogen clearance, and modulating the immune response to allergens. Created with BioRender.com.

Additionally, intraperitoneal injection of palmitic acid, a surfactant pre-cursor, labeled with tritium showed reduced [H3]-Sat-PC secretion in the BALF of STAT3 knockout mice, thus, demonstrating defects in lipid secretion and reduced de novo surfactant production [102]. Challenge via intratracheal injection of LPS demonstrated the deficiency in

surfactant protein production specific to SP-B and not SP-A, -C, or -D, which could indicate independent regulatory mechanisms in differing surfactant protein production [102]. However, the exact role of LIF in these processes requires further investigation.

### 7.2. Acute Inflammatory Response Regulation

Intratracheal LPS challenge in STAT3 knockout mice demonstrated a more sustained increase in macrophages, inflammatory cytokines (IL-1β, IL-6, and CXCL2), and increased alveolar damage [102]. Intratracheal injection of AV1-GFP, a non-proliferative adenovirus, to homozygous STAT3-depleted mice results in large airway enlargement secondary to loss of alveoli, bronchial epithelial cells sloughing, and focal pulmonary hemorrhage that was progressive throughout the 7 day experimental course [103]. Immunological comparison of bronchial epithelial cells revealed an 8-fold increase in caspase-3 positivity and a 4-fold increase in TUNEL positive stain uptake in homozygote STAT3 knockout mice compared to controls [103].

Utilizing a double-crossed, doxycycline-inducible expression of STAT3 transgenic mice model challenged with 95% oxygen, Lian et al. observed >50% survival in the STAT3 over-expression cohort at day 7 compared to 100% mortality observed in the control group [104]. RT-PCR performed on lung tissue demonstrated reduced mRNA expression of metalloproteases, specifically MMP-9 and MMP-12, before mortality in the STAT3 overexpressed cohort. This is correlated quantitatively with a 7-fold reduction in the number of alveolar neutrophils [104]. However, specific mechanisms in which STAT3, and possibly LIF, induces alteration in neutrophil trafficking in acute lung injury remain to be elucidated.

### 7.3. Innate Immunity Regulation

Experiments involving cytotrophoblast immune regulation of tissue macrophages during the invasion of uterine decidua shed light on the possible mechanism behind STAT3 modulation of innate immune cell trafficking and anti-inflammatory effects. IFN-γ released in response to inflammatory stress is a potent inducer of macrophages [105]. Specifically, in type 1 macrophages, IFN-γ stimulation mediated through STAT1 induces transcription of pro-inflammatory mediators such as TNF-α [105]. In 2014, Angham et al. identified LIF secreted from villous cytotrophoblast as a critical player in host tolerogenicity, allowing for implantation [106]. Their experiments on tissue macrophage co-stimulated with IFN-γ and LIF led to a 3-fold reduction in phosphorylated STAT1 compared to IFN-γ+/LIF- cells [79]. IFN-γ is known to reduce macrophage motility, which can be beneficial in the retention of activated macrophage at the target site via expression inhibition of chemotactic chemokine CCL2 and its receptor CCR2 [106]. Increased macrophage motility is observed in IFN-γ+/LIF+ co-stimulated tissue macrophages, which can support clearance of acute inflammatory response [106]. Furthermore, LIF stimulation leads to reduced TNF-α, CD14, and CD40 (antigen-presenting macrophages and DC surface antigen) expression, supporting reduced macrophage activation [106]. As mentioned earlier, LIF expression in macrophages shifts macrophages from a pro-inflammatory (M1) to an anti-inflammatory (M2) phenotype;, an M2 macrophages shift is observed in multiple pulmonary diseases [107] but the role of LIF in macrophage polarization in pulmonary diseases needs further investigation. Taken together, LIF functions in innate immune modulation by reducing macrophage activation, inflammasome production, and increasing clearance/motility, possibly through interference with IFN-γ-induced STAT1 phosphorylation [106].

### 7.4. Adaptive Immunity Regulation

Transgenic LIF overexpression in T cells results in the expansion of both CD4+/CD8+ lymphocytes and the B cells [108]. These mice also had hypergammaglobulinopathy with a marked reduction in thymic CD4+/CD8+ lymphocytes and an expansion of CD4+/CD8+ T cells in the peripheral lymph nodes [108]. Thymic parenchyma on gross pathology display loss of internal structure, suggesting that the observed lymphocytic clonal alteration might be secondary to disruption of stromal-lymphocyte immune synapses [108]. LIF can

promote adaptive immunity anergy by expansion of Foxp3+ cells from CD4+ T cells population [109]. However, whether LIF regulated T cell responses in pulmonary diseases requires investigation.

The role of the adaptive immune response in the pathogenesis of several pulmonary diseases is conflicting, such as in COPD and emphysema. When comparing the stage of the disease or during an exacerbation differing T cell profiles are observed, with a recent study showing reduced lymphocyte count observed in severe acute exacerbation of COPD that correlated with mortality and age [110]. Low lymphocyte count is also linked to poor exercise capacity and quality of life in COPD patients [111]. Alternatively, several studies suggest higher CD8+ T lymphocytes in COPD [112]. However, the neutrophil to lymphocyte ratio may be a better predictor for incidence of exacerbation and mortality in patients with COPD [113].

## 8. Conclusions

LIF biology is primarily studied in the contexts of stem cell biology, reproduction, and various cancers. However, recent studies suggest that LIF may be more multifunctional than we first thought, with LIF playing a role in immune responses, cell differentiation, and cell fate. Activation of multiple pathways by LIF, such as STAT3, gp130, and LIFR, needs to be investigated within the lungs in normal and diseased states. LIF is detected in epithelial and mesenchymal cells of the upper airways in human lung tissues and cultured cells when stimulated with proinflammatory stimuli [60]. Equally, LIFR is highly expressed in multiple lung cell types, especially type II pneumocytes, endothelial cells, and mesenchymal cells [84]. These cells are the main components of the alveolar-capillary barrier, which is a critical area of the lungs for gas exchange, fluid regulation, and immune infiltration [114]. Therefore, LIF and its receptors likely play critical roles in lung injury and repair. While this review has primarily focused on the role of LIF in immune responses, future approaches should also include LIF's potential ability to regulate the activity of various cell types involved in lung tissue remodeling, as LIF can regulate the expression of proteases (MMPs) [115] that influence the breakdown of the ECM, and LIF could play a major role in cell differentiation, proliferation, TGFβ and EMT signaling, and cell fate.

**Author Contributions:** Conceptualization, H.Y. and P.G.; writing-original draft preparation, H.Y. and P.G.; writing-review and editing, H.Y., S.Z., M.P., H.T., M.S., S.C., A.A., S.T. and P.G.; visualization, H.Y. and P.G. All authors have read and agreed to the published version of the manuscript.

**Funding:** This research received no external funding.

**Institutional Review Board Statement:** Not applicable.

**Informed Consent Statement:** Not applicable.

**Data Availability Statement:** No new data were created or analyzed in this study. Data sharing is not applicable to this article.

**Acknowledgments:** The authors would like to thank the Pulmonary Division of SUNY Downstate Health Sciences University and the Internal Medicine Residency Program for their support.

**Conflicts of Interest:** The authors declare no conflict of interest.

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
