# Peer review of "The Role of Leukemia Inhibitory Factor in Counteracting the Immunopathology of Acute and Chronic Lung Inflammatory Diseases"

_2673-527X, doi:10.3390/jor3020009_

Round 1

Reviewer 1 Report

1. The authors have provided a detailed narrative review on LIF with  particular focus on its possible  involvement in the rather neglected area of acute and chronic lung disease pathogenesis. It would be good to know what the authors perceive their intended audience to be. As it stands the review is pretty comprehensive and a good place for a potential researcher to get a good list of studies done and briefly what they say. What is more difficult is getting the overall message from each section, and it might improve its cohesive effectiveness if each section had a summary "take-away message(s)" sentence.

2. I feel that the authors have a rather old-fashioned view of diseases such as COPD and IPF. They are primarily NOT due to inflammation but due to epithelial activation and gene reprogramming leading to tissue  remodelling/fibrosis, in contrast to more acute events such a bacterial/viral pneumonia, ARDS and acute exacerbations in COPD/IPF etc. In such chronic disease TGFbeta is intimately involved, and so is EMT (full or partial).  Various allusions are made to TGF, SMADS, mTOR, Wnt etc which are intimately involved in remodelling/EMT activation, but no connections are made with such current insights into destructive pulmonary disease models.

3. On the same theme, on Line374 etc the authors boldly state that COPD is "driven" by adaptive immunity, quoting quite an old ref from 2004 (82), whose author did quite frequently make somewhat exuberant proclamations. Lymphocyte accumulation in airway tissue does occur in COPD but in very  advanced, end-stage disease, possible due to an auto-immune reaction to a lot of tissue injury or secondary to much chronic bacterial infection, but is unlikely to be a driver of earlier stage disease. The authors try and make a rather speculative connection between LIF and adaptive immunity, but it may be a bit if a straw man in COPD.

4. How are the authors saying LIF may be working in NSC lung cancer? This is very closely related to COPD epidemiologically, and EMT seems important in its invasiveness and metastatic aggressiveness. Could LIF hook into this?

5. Section 5.4: the first and second halves don`t seem to bind together coherently: the former says that Type 2 pneumocytes derive from BM cells, while the latter suddenly shifts to emphasis on BM cell-derived NFkappaB stimulation of epithelial cells to produce LIF. How do these data fit together; are they both likely true or are they contrasting theories?

6. Section 7.5 on macrophages: there is a phenotypic change in macrophage pattern in smoker/COPD airway lumen, with more M2 rather than M1 cells. Could LIF be responsible on the basis of the data described? 

Overall, English is pretty good, although the style of relentlessly piling studies one on top of the other is a bit gruelling.

There are a few typos , e.g. Line265, it should be "from" COPD patients and not "of". Line 290: this sentence is garbled and does not seem to mean anything. Section 7.4, the first sentence does not have a verb.

Author Response

  1. The authors have provided a detailed narrative review on LIF with particular focus on its possible involvement in the rather neglected area of acute and chronic lung disease pathogenesis. It would be good to know what the authors perceive their intended audience to be. As it stands the review is pretty comprehensive and a good place for a potential researcher to get a good list of studies done and briefly what they say. What is more difficult is getting the overall message from each section, and it might improve its cohesive effectiveness if each section had a summary "take-away message(s)" sentence.

Response: We thank the reviewer for their constructive and helpful comments. We wrote this paper for physicians and scientists with an interest in pulmonary immunology as an intended audience. We have made modifications to better summarize each section. We have also tried to incorporate other possible LIF functions beyond inflammation. However, most of LIF’s functions explored in the pulmonary field are very inflammation-focused, and discussing other possible functions may be deemed as a perspective point of view rather than a literature review.

  1. I feel that the authors have a rather old-fashioned view of diseases such as COPD and IPF. They are primarily NOT due to inflammation but due to epithelial activation and gene reprogramming leading to tissue remodelling/fibrosis, in contrast to more acute events such a bacterial/viral pneumonia, ARDS and acute exacerbations in COPD/IPF etc. In such chronic disease TGFbeta is intimately involved, and so is EMT (full or partial). Various allusions are made to TGF, SMADS, mTOR, Wnt, etc. which are intimately involved in remodelling/EMT activation, but no connections are made with such current insights into destructive pulmonary disease models.

Response: We do appreciate the reviewers’ comments on other possible factors driving pulmonary diseases and we have tried to incorporate this into the manuscript. But I do not think that epithelial activation, EMT or TGFbeta signaling are new views or novel pathways within most of these diseases. However, we agree that inflammation is not the only factor in these diseases and have modified the manuscript to avoid any miscommunication. We have added several sentences to outline that many players are contributing to pulmonary diseases and no signaling pathway is fully driving diseases initiation or progression. We have also added two sections on IPF and asthma in section 6. Please see lines 323-352.

  1. On the same theme, on Line374 etc the authors boldly state that COPD is "driven" by adaptive immunity, quoting quite an old ref from 2004 (82), whose author did quite frequently make somewhat exuberant proclamations. Lymphocyte accumulation in airway tissue does occur in COPD but in very advanced, end-stage disease, possible due to an auto-immune reaction to a lot of tissue injury or secondary to much chronic bacterial infection, but is unlikely to be a driver of earlier stage disease. The authors try and make a rather speculative connection between LIF and adaptive immunity, but it may be a bit if a straw man in COPD.

Response: We agree that the sentence on line 374 was poorly presented and have modified this and other sections to better reflect the current views on this disease. We have removed Dr. Barnes 2004 paper from our references

  1. How are the authors saying LIF may be working in NSC lung cancer? This is very closely related to COPD epidemiologically, and EMT seems important in its invasiveness and metastatic aggressiveness. Could LIF hook into this?

Response: We agree with the reviewer and have expanded this section. Please see lines 262-284.

  1. Section 5.4: the first and second halves don`t seem to bind together coherently: the former says that Type 2 pneumocytes derive from BM cells, while the latter suddenly shifts to emphasis on BM cell-derived NFkappaB stimulation of epithelial cells to produce LIF. How do these data fit together; are they both likely true or are they contrasting theories?

Response: We agree with the reviewer about the confusion in this section and have edited this section to remove the confusion. Please see lines 437-457.

  1. Section 7.5 on macrophages: there is a phenotypic change in macrophage pattern in smoker/COPD airway lumen, with more M2 rather than M1 cells. Could LIF be responsible on the basis of the data described?

Response: We have updated section 7.3 with the following line on M2 Macrophages: “As mentioned already, LIF expression in macrophages shift macrophages from a pro-inflammatory (M1) to an anti-inflammatory (M2) phenotype [74] and an M2 macrophages shift is observed in multiple pulmonary diseases [104] but the role of LIF in macrophage polarization in pulmonary diseases needs further investigation.” We have also added several lines to the cancer section. Please see lines 323-352.

Comments on the Quality of English Language

Overall, English is pretty good, although the style of relentlessly piling studies one on top of the other is a bit gruelling.

Response: We have tried to make the topic easier to read but adding more summary sentences

There are a few typos , e.g. Line265, it should be "from" COPD patients and not "of". Line 290: this sentence is garbled and does not seem to mean anything. Section 7.4, the first sentence does not have a verb.

Response: We have made these changes

Reviewer 2 Report

Leukaemia Inhibitory Factor (LIF) is a multi-functional polypeptide cytokine/growth factor and it is one of the most studied ligand–receptor complexes, However, little is known about this cytokine in the context of normal lung function or indeed, inflammation. In the paper, " The Role of Leukemia Inhibitory Factor in Counteracting the Immunopathology of Acute and Chronic Lung Inflammatory Diseases", Dr. Howard Yu and his colleagues review the recently studies about the feature of LIF signaling  and its role in airway inflammation.  

Major comments: No major concern

Minor comments:

Line 284:  change ARDs  to ARDS

Line 317 Figure 1 (gp120) and Line 320 (Gp120): change gp120 to gp130

Author Response

Leukaemia Inhibitory Factor (LIF) is a multi-functional polypeptide cytokine/growth factor and it is one of the most studied ligand–receptor complexes, However, little is known about this cytokine in the context of normal lung function or indeed, inflammation. In the paper, " The Role of Leukemia Inhibitory Factor in Counteracting the Immunopathology of Acute and Chronic Lung Inflammatory Diseases", Dr. Howard Yu and his colleagues review the recently studies about the feature of LIF signaling  and its role in airway inflammation. 

Response: We thank the reviewer for their constructive and helpful comments

Minor comments:

Line 284:  change ARDs  to ARDS

Line 317 Figure 1 (gp120) and Line 320 (Gp120): change gp120 to gp130

Response: We have made these changes

Round 2

Reviewer 1 Report

1. The authors have undertaken a fairly extensive modification of the manuscript, taking to account most of my comments. I think a specific brief summary of each section would be good, saying what your 1-2 main take-home messages are. 

2. Lines 161-66: the authors seem reluctant to use the word "remodelling" as fundamental to respiratory disease processes; but would fit in well after "repair and regeneration" which do not of course cause dysfunction. Later TGF and EMT get mentioned (L334) but their central importance in fibrotic disease states in the lung, secondary to epithelial activation,  should not be so marginalised.  This would also fit well into the last sentence of the Conclusions.

3. Line 36: should read "regarded as playing" , and might well be excised.

nil more

Author Response

Reviewer 1:

  1. The authors have undertaken a fairly extensive modification of the manuscript, taking to account most of my comments. I think a specific brief summary of each section would be good, saying what your 1-2 main take-home messages are.

Response: We have added brief summary sentences to several sections. Please see the track changes in the sections needing a summary sentence

  1. Lines 161-66: the authors seem reluctant to use the word "remodelling" as fundamental to respiratory disease processes; but would fit in well after "repair and regeneration" which do not of course cause dysfunction. Later TGF and EMT get mentioned (L334) but their central importance in fibrotic disease states in the lung, secondary to epithelial activation, should not be so marginalised. This would also fit well into the last sentence of the Conclusions.

Response: We have made changes to address these suggestions. We are not reluctant to use these terms for lung diseases but the exact role of LIF in tissue remodeling and TGFbeta/EMT signaling in the lungs is not clear. As this is a LIF focus manuscript, we want to keep (as much as possible) the known LIF responses in lung diseases, rather than a review of possible players in lung diseases or only fibrotic disease states. Successful repair and regeneration do not cause lung dysfunction but we must also be mindful that remodeling associated with lung disease is linked to dysfunctional repair and regeneration. Also, we previously expanded on TGF beta and EMT in the last version of the manuscript in the NSCLC section and we did mention the importance of it in other pulmonary diseases (lines 280-285).

Please also see lines 104-106, 167-170, 228-230, 282, and 475-476.

  1. Line 36: should read "regarded as playing" , and might well be excised.

Response: We have made this change. Please see line 37.